# Morphine exposure exacerbates HIV-1 Tat driven changes to neuroinflammatory factors in cultured astrocytes

Kenneth Chen[1], Thienlong Phan[1], Angel Lin[1], Luca Sardo[1,2], Anthony R. Mele[3,4], Michael R. Nonnemacher[3,4], Zachary Klase[1]*

1 Department of Biological Sciences, University of the Sciences, Philadelphia, Pennsylvania, United States of America, 2 Current institution – Department of Infectious Diseases and Vaccines, MRL, Merck & Co., Inc., West Point, Pennsylvania, United States of America, 3 Department of Microbiology and Immunology, Drexel University College of Medicine, Philadelphia, Pennsylvania, United States of America, 4 Center for Molecular Virology and Translational Neuroscience, Institute for Molecular Medicine and Infectious Disease, Drexel University College of Medicine, Philadelphia, Pennsylvania, United States of America

* z.klase@usciences.edu

**Data Availability Statement:** All relevant data are within the manuscript and its Supporting Information files.

## Abstract

Despite antiretroviral therapy human immunodeficiency virus type-1 (HIV-1) infection results in neuroinflammation of the central nervous system that can cause HIV-associated neuro-cognitive disorders (HAND). The molecular mechanisms involved in the development of HAND are unclear, however, they are likely due to both direct and indirect consequences of HIV-1 infection and inflammation of the central nervous system. Additionally, opioid abuse in infected individuals has the potential to exacerbate HIV-comorbidities, such as HAND. Although restricted for productive HIV replication, astrocytes (comprising 40–70% of all brain cells) likely play a significant role in neuropathogenesis in infected individuals due to the production and response of viral proteins. The HIV-1 protein Tat is critical for viral transcription, causes neuroinflammation, and can be secreted from infected cells to affect uninfected bystander cells. The Wnt/β-catenin signaling cascade plays an integral role in restricting HIV-1 infection in part by negatively regulating HIV-1 Tat function. Conversely, Tat can overcome this negative regulation and inhibit β-catenin signaling by sequestering the critical transcription factor TCF-4 from binding to β-catenin. Here, we aimed to explore how opiate exposure affects Tat-mediated suppression of β-catenin in astrocytes and the downstream modulation of neuroinflammatory genes. We observed that morphine can potentiate Tat suppression of β-catenin activity in human astrocytes. In contrast, Tat mutants deficient in secretion, and lacking neurotoxic effects, do not affect β-catenin activity in the presence or absence of morphine. Finally, morphine treatment of astrocytes was sufficient to reduce the expression of genes involved in neuroinflammation. Examining the molecular mechanisms of how HIV-1 infection and opiate exposure exacerbate neuroinflammation may help us inform or predict disease progression prior to HAND development.

**Funding:** This work was funded by the Public Health Service, National Institutes of Health, through grants from the (1) National Institute of Drug Abuse (NIDA) DP2 DA044550 (PI, Zachary A. Klase) and R03 DA039733 (PI, Zachary A. Klase); (2) the National Institute of Neurological Disorders and Stroke (NINDS) R01 NS089435 (PI, Michael R. Nonnemacher); and (3) the Ruth L. Kirschstein National Research Service Award T32 MH079785 (Dr. Brian Wigdahl, Principal Investigator of the Drexel University College of Medicine component and Dr. Olimpia Meucci as Co-Director). The funders had no role in study design, data collection and analysis, decision to publish, or preparation of the manuscript.

**Competing interests:** The authors have declared that no competing interests exist.

## Introduction

Human immunodeficiency virus type 1 (HIV-1) is the etiological agent of acquired immuno-deficiency syndrome (AIDS). Without intervention, HIV-1-infected individuals become increasingly susceptible to opportunistic infections due to progressive decline in their CD4 + T-cell population, which ultimately results in death. Antiretroviral therapy (ART), however, is highly effective at managing HIV-1 viral replication, preventing subsequent infection, and preserving immune function. Despite the effectiveness of ART, HIV-1-infected cells persist and can be found in the periphery and in various organ systems throughout the body [1–3]. In particular, HIV-1 dissemination in the central nervous system (CNS) occurs early in infection during the acute stage, with HIV-1 RNA also detectable in the cerebrospinal fluid (CSF) during the chronic infection stage [4–9]. Regardless of adherence to ART, viral residency in the CNS establishes long-term low levels of inflammation that contribute, in part, to the development of HIV-associated neurocognitive disorders (HAND), which lead to significant burdens on diagnostics, treatment, and quality of life [10–13]. Understanding the molecular mechanisms of HIV-1 disease development during chronic HIV-1 infection is critical for the rigor required for further developing therapeutic interventions.

The mechanism of HIV-1 dissemination throughout the CNS is still largely unknown. The virus itself may cross the blood brain barrier (BBB) or may be introduced due to trafficking of infected CD4+ T-cells from the periphery [14]. CNS-resident, long-lived cell types such as monocytes, macrophages, and astrocytes have all been implicated in HIV-1 infection and neu-roinflammation, though the establishment of latent and prolonged infection in these cell types within the CNS remains controversial [15]. Astrocytes, specifically, can be permissive to HIV-1 infection, but these cells are restrictive to productive replication, and there is limited data demonstrating infection *in vivo* [16–18]. Within the CNS, astrocytes maintain the homeostatic balance and play key roles in mediating cerebral blood flow by maintaining the BBB, providing neuronal support through neurotransmitter processing, regulating nutrients and growth factors, and modulating inflammation [19–22]. HIV-1 does not infect neurons, implying that development of HAND might be mediated indirectly, for example through astrocyte dysregu-lation and inflammation. A measurable percentage of astrocytes in the CNS can harbor HIV-1 integrated DNA, making astrocytes a likely contributor in the development of HAND [23–26].

In addition to cytotoxic effects due to direct infection with HIV-1, the CNS represents an environment where secreted viral proteins, cytokines, chemokines, and small molecules can all contribute to neurotoxicity [4, 27–31]. Therefore, proximity to HIV-1 infected cells can result in cell and tissue damage. In particular, the HIV-1 viral protein Tat has been associated with clinical neuropathogenicity in the CNS, specifically through induction of inflammatory pro-cesses including astrogliosis [32–40]. HIV-1 Tat has classically been described in its function as the viral transactivator for HIV-1 transcription in infected cells [41]. HIV-1 transcription is mediated by Tat binding to the transactivation response (TAR) element, an RNA stem-loop encoded by the HIV-1 long-terminal repeat (LTR) promoter, which results in the recruitment of the host positive transcription elongation factor b (p-TEFb) complex, which phosphorylates RNA pol II, thereby driving HIV-1 transcriptional elongation [41–44].

HIV-1 is subject to genetic variation, which may alter the functionality of the encoded viral proteins. *In vitro*, Tat expressing cells have been demonstrated to readily secrete Tat extracellu-larly, a function mediated by a tryptophan at amino acid position 11 (W11), as well as residues 49–51 within the basic domain [41, 45–47]. Tat variants containing mutations at this position, such as W11F or W11Y, are deficient in both secretion from Tat-producing cells and cyto-plasmic translocation in recipient cells [46, 48]. In the CNS, Tat can also be secreted from HIV-1 infected cells, such as macrophages and astrocytes, which results in direct neurotoxicity;

therefore, Tat secretion-deficient mutants may alleviate this phenotype within patients [49–55]. Finally, mutations at a conserved cysteine at amino acid position 31, C31, which is within the cysteine-rich domain, have been shown to abrogate synaptodendritic injury in neurons and normalize levels of amyloid beta production, therefore conferring protection from Tat-mediated neurotoxicity [41, 56–58].

The Wnt/ β-catenin pathway has been implicated in suppressing replication of HIV-1 in multiple cell types, including astrocytes, and conversely, downregulation of Wnt/β-catenin signaling by inflammatory cytokines promotes HIV-1 replication [59–63]. Interestingly, HIV-1 Tat has also been shown to inhibit Wnt/β-catenin signaling in astrocytes, therefore alleviating, in part, the negative regulation of Tat-mediated transactivation of the LTR by β-catenin [60, 63, 64]. β-catenin is a central mediator of cell adhesion and transcriptional activity within the nucleus. Wnt ligands are derived from small glycoproteins and bind to seven-transmembrane frizzled receptors and low-density lipoprotein receptor related protein 5 and 6 (LRP5/6) co-receptors to transduce a signal leading to removal of phosphate groups and increased downstream stability of β-catenin. When bound to members of the T-cell factor/lymphoid enhancer factor (TCF/LEF) family of transcription factors, β-catenin can modulate the activity of hundreds of target genes. Compared to other cell types and despite robust levels of signaling, less is known about the role of Wnt/ β-catenin in astrocyte function [59, 62, 65]. Approximately 150 β-catenin-regulated genes are estimated in primary astrocytes are comprising at least five broad categories including inflammation, transport, exocytosis, apoptosis, and trafficking [61]. Of these, β-catenin responsive genes that are neuroinflammatory in nature include *BDNF* (Brain-Derived Neurotrophic Factor), a member of the BDNF/*TrkB* (Tropomyosin receptor kinase B) signaling axis [66–68].

In addition to neurotoxicity associated with HIV-1 infection, mounting evidence suggests exacerbation of these effects in the presence of opiates [69–72]. Illicit use and drug abuse contribute significantly to HIV-1 infections and transmission [73], therefore elucidating the interaction between the cells of the CNS, opiates, and HIV-1 Tat is necessary to understanding HAND disease progression. The mu opioid receptor is the most common target of opiates and is expressed on many different cell types; morphine, a derivative of heroin, binds to this receptor at a very high affinity. The effects of Tat and morphine on cells of the CNS have been explored extensively *in vitro* and *in vivo*, and phenotypes observed could correlate with symptoms of neurodegeneration [30]. For example, *in vitro* expression of Tat, exposure to morphine, or both has been demonstrated to decrease fidelity of the BBB and subsequently increase immune cells transmigration across the barrier *in vitro* [74]. Additionally, *in vivo* Tat expression or morphine exposure in transgenic mice can decrease the proliferation of cells of the CNS, as well as affecting reaction times and performance testing [75, 76].

Given the established importance of Tat-mediated suppression of Wnt/β-catenin signaling in HIV-1 infected astrocytes, here we explored the compounding effects of morphine exposure and HIV-1 Tat expression on β-catenin signaling in human astrocytes. We confirmed previously reported results by our laboratory and others that Tat expression is sufficient to downregulate β-catenin activity, as measured from a β-catenin promoter reporter, in both an astrocyte cell line (U87MG) and in primary fetal astrocytes (PFAs). Interestingly, the presence of morphine potentiates the effects of Tat on suppression of β-catenin activity. We also found that morphine exposure, Tat expression, or a combination of the two are able to decrease the active form of β-catenin protein in PFAs. Despite donor to donor variability, we observe recovery of β-catenin activity in the presence of Tat mutants W11F, W11Y, and C31R compared to WT Tat in PFAs in the presence or absence of morphine. Finally, we observed loss of expression of genes associated with neuroinflammation in the presence of morphine in U87MG cells and PFAs. Taken together, these results suggest a compounding effect of morphine exposure on β-catenin activity in astrocytes which could affect neuroinflammation and/or neurotoxicity in HIV-1 infected individuals.

## Materials and methods

### Cells

U87MG cells (ATCC) were maintained in Eagles Minimal Essential Media (EMEM) supplemented with 10% Fetal Bovine Serum (FBS), 4.5 g/L of L-glutamine, and 1% penicillin/streptomycin. Primary fetal astrocytes (PFAs) were maintained in Astrocyte Growth Media composed of Dulbecco's Modified Eagle Media/Ham's F-12 (DMEM/F-12) supplemented with 15% FBS (exosome free), 4.5 g/L of L-glutamine, 100 ug/mL gentamicin, 10 ug/mL Amphotericin B, and 20 ug/mL insulin. PFAs were obtained from the Temple University Comprehensive NeuroAIDS Center (CNAC), Basic Science Core I according to standard procedure. In brief, Fetal brain tissue (gestational age, 16–18 weeks) was obtained from elective abortion procedures performed in full compliance with National Institutes of Health and Temple University ethical guidelines. The tissue was washed with cold Hanks balanced salt solution (HBSS), meninges and blood vessels were removed. For glial cultures, tissue in HBSS was digested with 0.25% trypsin for 30 min at 37°C. Trypsin was neutralized with FBA, and the tissue was further dissociated to obtain single-cell suspensions. For glial cultures, cells were plated in mixed glial growth media (same as Astrocyte Growth Media, but with 10% FBS). The mixed culture was maintained under 10% $CO_2$ for 5 days, and the medium was fully replaced to remove any cell debris. To enrich for microglia and astrocytes, flasks were placed on an orbital shaker for 14–18 hours at 200 rpm in growth media. Detached cells constitute the microglial component of the culture and were collected and plated into a new flask containing microglial media. Monolayers that remain after shaking constitute the astrocytes and are fed with astrocyte media. Purity of cell type specific cultures was assessed by immunolabeling with anti-GFAP and–GLAST1 for astrocytes, -lba-1 and–CD11b for microglial and–MAP2 or–neurofilament for neurons. Three individual PFA donors are included in this study (Donor #1, Donor #2, Donor #3) and are used in experiments where indicated. Cells were cultured at 37°C in 5% $CO_2$.

### Plasmids and transfections

U87MG cells or PFAs were seeded in 6, 12 or 24 well plates at a density designed to yield 90% confluency within 24 hours. Cells were transfected using Transfectin (Bio-Rad) or Lipofectamine 3000 (Invitrogen) according to the manufacturer's instructions. pUC19 plasmid was used to normalize total transfection input DNA across all conditions. pmaxGFP plasmid (Lonza) was included to monitor transfection efficiency by fluorescence microscopy. M50 Super 8x TOPFlash β-catenin plasmid [77] (a gift from Randall Moon (Addgene plasmid # 12456; http://n2t.net/addgene:12456; RRID:Addgene_12456) and the Cignal TCF/LEF β-catenin plasmid (Qiagen #336841) are luciferase reporter constructs under the control of β-catenin-mediated transcriptional activation. Flag-Tat$_{86}$WT and Flag-Tat$_{101}$WT are HIV-1 Tat expression vectors that have been previously described [78, 79]. Flag-Tat$_{101}$WT was used to generate Tat mutants, Tat$_{101}$C31R, Tat$_{101}$W11F, and Tat$_{101}$W11Y utilizing the GeneArt Site-Directed Mutagenesis System (Invitrogen).

### β-catenin activation and drug treatment

Where indicated, 24 hours post-transfection, at 90% transfection efficiency, cells were treated with the β-catenin activator Lithium Chloride [(LiCl); 50mM; Sigma] in the presence or absence of Morphine sulfate pentahydrate (50nM or 500nM; Sigma). Alternatively, where indicated, 24 hours post-transfection, cells were treated with the β-catenin activator (2'Z,3'E)-6-Bromoindirubin-3′-oxime [(BIO); 1mM; Sigma] in the presence or absence of morphine (500nM).

## β-catenin luciferase assay

24 hours post-transfection or post-treatment, where indicated, cells were lysed with passive lysis buffer (Promega). Cell lysates were incubated in equal volumes with the DualGlo luciferase reagent (Promega) and luciferase was measured in a luminometer. Data shown represent the average of technical triplicates.

## SDS-PAGE and Western blotting

24 hours post-treatment, cells were lysed in RIPA buffer (ThermoFisher) with Halt™ Protease inhibitor Cocktail (ThermoFisher) and incubated under agitation at 4°C. Lysates were briefly homogenized by syringe and debris were cleared by centrifugation. Cell lysates were incubated in equal volume Bolt™ Sample Reducing Agent (Invitrogen) and Bolt™ LDS sample buffer (Invitrogen) for 10min at 98°C. Cell lysates were separated on Bolt™ 4–12% Bis-Tris plus gels (ThermoFisher) at 200 volts. Transfer of proteins to 0.2um pore size PVDF membranes was performed using Mini Blot Module (Invitrogen) transfer cassettes and Bolt™ transfer buffer made with 10% methanol and Bolt™ antioxidant (Invitrogen). Proteins were transferred at 20 volts for 1 hour. Membranes were blocked in 5% milk in PBS with 0.1% Nonidet™ P-40 (NP-40) detergent at 25°C for 1 hour, stained with primary antibodies at 4°C overnight, washed, and stained with secondary antibodies at 25°C for 1 hour. The α-total β-catenin antibody (ThermoFisher, cat #MA1-301) and the α-active β-catenin antibody (US Biological Life Sciences, cat# C2069-47) were used at a 1:1000 dilution. The antibody against total β-catenin recognizes all forms of the protein, however, the antibody against active β-catenin recognizes unphosphorylated Ser32 and Ser37. The α-GAPDH antibody (Abcam, cat# ab9485) was used at a 1:3000 dilution. Membrane chemiluminescent signals were imaged with BioRad imager and Quantity One software. Images were analyzed, processed and quantitated using ImageJ software. The mean gray scale values for total and active β-catenin were normalized to GAPDH.

For Tat Western blotting a cocktail of antibodies was used. Membranes were blocked in 5% milk + PBS-T (PBS + 0.05% Tween-20) and incubated with a cocktail of primary antibodies at 4°C overnight: α-Tat (1:1000; Abcam #ab63957), α-Tat (1:1000; #7377 NIH AIDS Reagent Program, Division of AIDS, NIAID, NIH: Anti-HIV-1 HXB2 Tat Monoclonal (1D9) from Dr. Dag E. Helland [80]), and α-Flag (1:5000; Invitrogen #MA1-142). Alternatively, membranes were incubated with α-β-Actin (1:5000; Sigma #A1978). Membranes were washed 1x with PBS-T and incubated with secondary antibodies at 25°C for 2hrs (Goat α-Mouse; 1:5000; BioRad #170–6516).

## qRT-PCR

48 hours post-treatment, RNA was harvested using TRIzol (Invitrogen) reagent according to manufacturer's *instructions*. RNA concentration was quantified by Nanodrop (ThermoFisher).

Reverse transcription was performed using the SuperScript III RT kit (ThermoFisher) using 1 μg of input RNA. The cDNA samples were diluted 1:20 and 1μl of sample template was added to a reaction mix containing 2.5μl of SYBR® green mix, 1.5μl of primer (5μM stock; Table 1) and 1μl of RNase-free water. The qPCR reaction was run using under the following conditions:

95°C—10 min

95°C—15 sec x39

$T_A$—30 sec

72°C—30 sec

**Table 1. mRNA primer set.**

| mRNA primer | Sequence | Length | $T_M$ (˚C) | $T_A$ (˚C) |
|---|---|---|---|---|
| NLRP1 For | TGG CAC ATC CTA GGG AAA TC | 20 | 54.4 | 53.8 |
| NLRP1 Rev | TCC TCA CGT GAC AGC AGA AC | 20 | 57.1 | |
| BDNF For | GAG CCC TGT ATC AAC CCA GA | 20 | 56.4 | 53.3 |
| BDNF Rev | CAA TGC CAA CTC CAC ATA GC | 20 | 54.1 | |
| TrkB For | GGG ACA CCA CGA ACA GAA GT | 20 | 57.2 | 56.1 |
| TrkB Rev | GAC GCA ATC ACC ACC ACA G | 19 | 56.5 | |
| GAPDH For | AGA AGG CTG GGG CTC ATT TG | 20 | 57.8 | 57.6 |
| GAPDH Rev | AGG GGC CAT CCA CAG TCT TC | 20 | 59.4 | |

## Statistics and data analysis

Data from biological or technical triplicates in experiments were statistically analyzed in GraphPad PRISM using two-tailed unpaired Student's *t* test. Where indicated, data were statistically analyzed in GraphPad PRISM using two-way ANOVA followed by Tukey's multiple comparisons test. P-values were calculated as significant as follows: *, p ≤ 0.05; **, p ≤ 0.01; ***, p ≤ 0.001; ****, p ≤ 0.0001.

## Results

### HIV-1 Tat expression suppresses β-catenin activity in astrocytes

We have recently determined that HIV-1 Tat-mediated suppression of specific miRNAs affects downstream signaling from the Wnt/β-catenin pathway [81]. The modulation of β-catenin activity by Tat could alleviate repression of basal HIV-1 transcription, resulting in enhanced neuroinflammation in the CNS [60, 63, 64]. To confirm our previous results and to address the ability of Tat to modulate β-catenin activity in physiologically relevant cells, we utilized the U87MG astrocytic cell line and primary fetal astrocyte (PFA) donors in our study. We co-transfected these cells with increasing concentrations of a full length, wildtype Tat ($Tat_{101}WT$) expression construct and a β-catenin responsive luciferase reporter construct in the presence of LiCl to stimulate β-catenin activity (Fig 1). LiCl acts as an agonist of the β-catenin signaling pathway by inhibiting GSK3β activity and stabilizing uninhibited β-catenin [82]. Tat expression was confirmed by Western Blot analysis (S1 Fig) and LTR-Luciferase reporter (Data not shown). In both U87MGs (Fig 1A) and PFAs (Fig 1B), we observed Tat-mediated suppression of β-catenin signaling. At the highest concentration of Tat, we observed a greater than 4-fold suppression in U87MG cells (p = 0.0011) and a 3-fold suppression in PFAs compared to the no Tat control (p = 0.0027). These data demonstrate that Tat can suppress β-catenin activity in both immortalized and primary astrocytes.

### Morphine treatment potentiates Tat effects on β-catenin activity in astrocytes

Given the correlates between substance abuse and HIV-1 infection, we next wanted to determine the combined effect of morphine and Tat expression on β-catenin signaling in U87MGs and PFAs. We again co-transfected these cells with increasing concentrations of $Tat_{101}WT$ and the β-catenin luciferase reporter construct in the presence of LiCl. One day post-transfection, we treated cells with or without increasing concentrations of morphine (U87MGs: 50nM, 500nM; PFAs: 500nM); morphine concentrations were selected based on literature describing optimal stimulation of cells with minimal toxicity [83–85]. In the absence of Tat, morphine

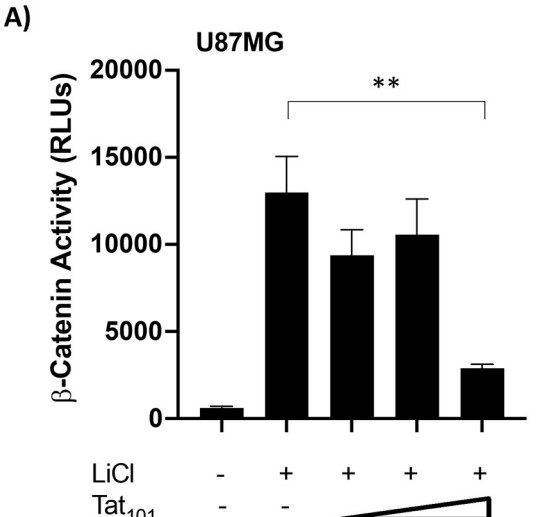

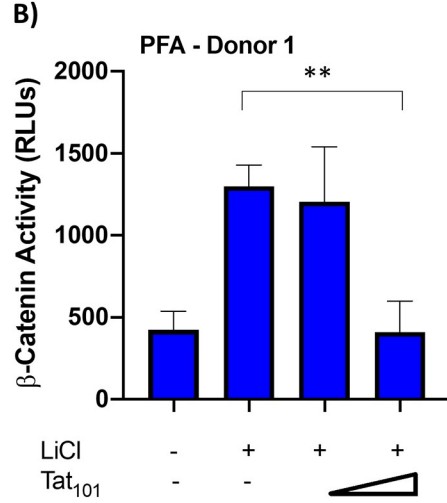

**Fig 1. Tat$_{101}$WT suppressed β-catenin signaling activity in U87MG astrocytes and primary fetal astrocytes (PFAs).** U87MGs (A) and PFAs from Donor 1 (B) were co-transfected with the TCF/LEF β-catenin luciferase reporter plasmid and increasing concentrations of the Flag-Tat$_{101}$ WT expression vector [A) 0.35ng, 3.5ng, 35ng; B) 3.5ng, 35ng] 24 hours post-transfection, cells were treated with LiCl (50mM), a β-catenin activator. Luciferase activity was measured 24 hours post-treatment. Data shown is the average of technical triplicates from one experiment ± SD. ** $p<0.01$ by two-tailed student's t test.

treatment (500nM) resulted in a small, but non-significant suppression of β-catenin activity in U87MGs (Fig 2A) and PFAs (Fig 2B). At low concentrations of Tat, morphine treatment (50nM in U87MGs and 500nM in PFAs) results in a further reduction of β-catenin signaling compared to the untreated control. At high concentrations of Tat, however, we observed little to no effect of morphine treatment on β-catenin signaling due to efficient suppression of Tat alone.

We verified the effect of Tat and morphine on β-catenin suppression in PFAs, by measuring the levels of active (unphosphorylated at Ser37 and Thr41) and total β-catenin proteins using antibodies specific for the two forms of the protein (Fig 3). Compared to untreated cells,

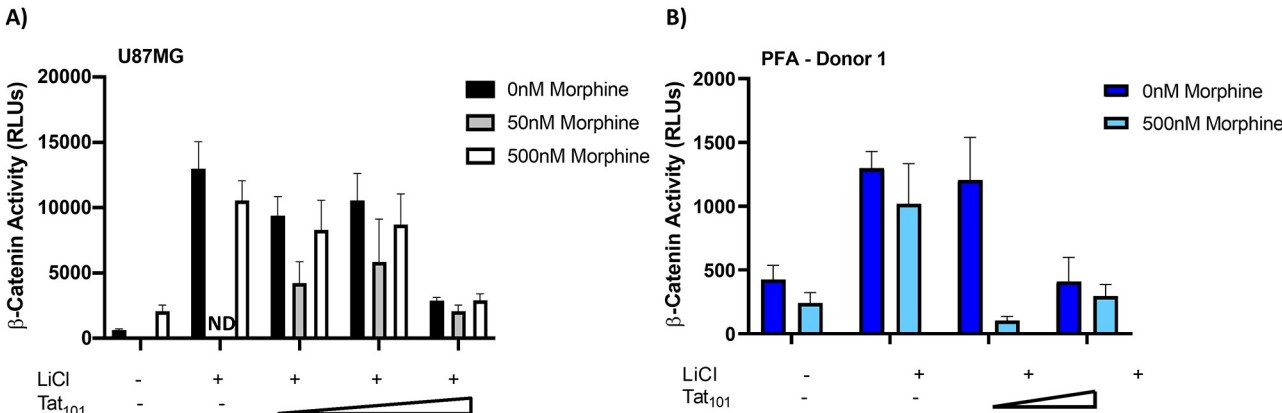

**Fig 2. Morphine potentiates Tat mediated suppression of β-catenin signaling.** U87MGs (A) and PFAs from Donor 1 (B) were co-transfected with the TCF/LEF β-catenin luciferase reporter plasmid and increasing concentrations of the Flag-Tat$_{101}$ WT expression vector [A) 0.35ng, 3.5ng, 35ng; B) 3.5ng, 35ng]. 24 hours post-transfection, cells were treated with LiCl (50mM) in the presence or absence of Morphine (50nM or 500nM), as indicated. Luciferase activity was measured 24 hours post-treatment. Untreated control data from Fig 1 is replotted here for reference. Data shown is the average of technical triplicates from one experiment ± SD. ** $p<0.01$ by two-tailed student's t test.

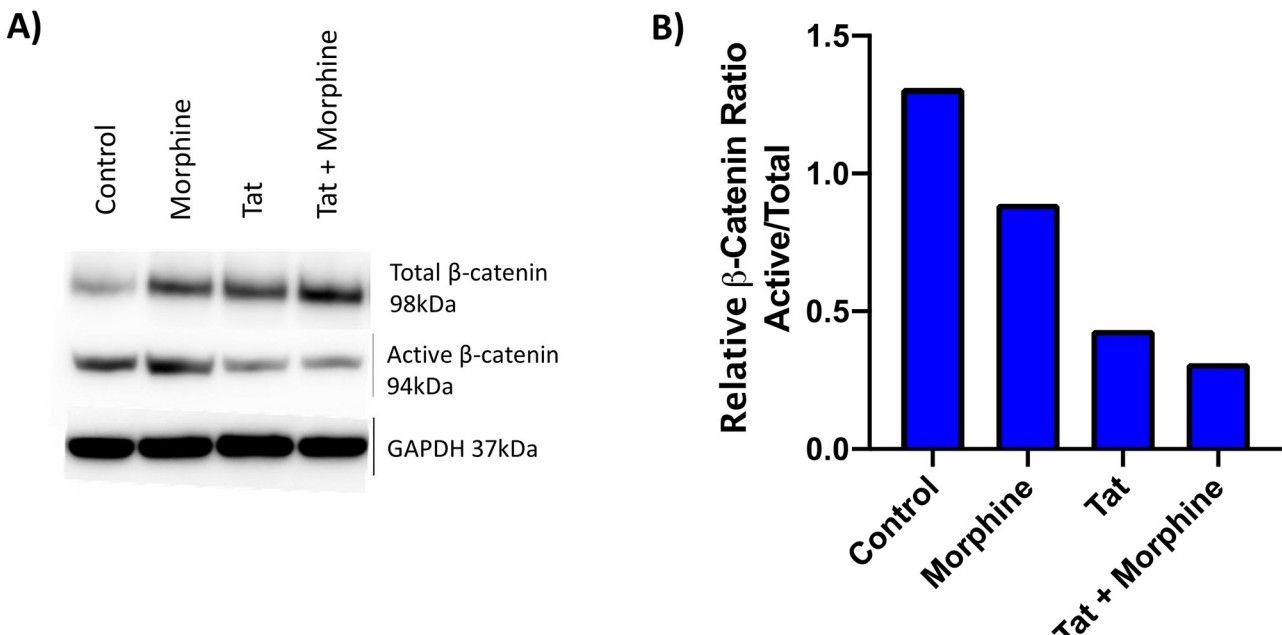

**Fig 3. Expression of HIV-1 Tat in PFAs results in loss of active β-catenin protein levels.** PFAs (Donor #1) were transfected with empty vector (pUC19) or the Flag-Tat$_{101}$WT (0.35ng) expression vector. 24 hours post-transfection, cells were treated with or without Morphine (500nM). Cells were collected and lysed for SDS-PAGE and western blotting 24 hours post-treatment (A) Proteins extracts were immunoblotted for total β-catenin, active β-catenin, and GAPDH. (B) Densitometry was performed on the resulting immunoblot and used to determine the ratio of active to total β-catenin for each condition. Data shown is from one experiment.

morphine increased total β-catenin levels but did not change active β-catenin levels, resulting in a decrease in the ratio of active to total β-catenin (~1.5 fold; Fig 3B). In contrast, Tat expression (low) in PFAs lowered active β-catenin levels compared to the untransfected control, resulting in a larger decrease in the ratio of active to total β-catenin compared to morphine alone (~2 fold; Fig 3B). Finally, a combination of Tat expression and morphine treatment further reduced the ratio of active to total β-catenin. Taken together, these results suggest that the observed loss of β-catenin activity with Tat expression and/or morphine treatment correlates with a loss of the active form of β-catenin protein in astrocytes.

## Donor-dependent effects of WT Tat and Tat mutants on β-catenin suppression in PFAs

Here, we have demonstrated that HIV-1 Tat can suppress β-catenin activation, however, to further define this phenotype, we explored the connection to Tat function. Tat is encoded by HIV-1 on two exons, the first of which is sufficient for the primary function of Tat to act as the viral transactivator and drive the transcriptional elongation of viral genes [86–88]. The functions of Tat encoded by the second exon contribute to viral replication and the modulation of interferon-stimulated genes in antigen presenting cells [89, 90]. Widely used laboratory-adapted strains of HIV-1 produce a truncated form of Tat, encoded by the first exon plus 14 amino acids of exon 2, Tat$_{86}$. The full-length transcript, however, encodes for the full-length form of Tat, Tat$_{101}$. To determine if the two different Tat variants have an effect on suppression of β-catenin activity, we co-transfected two PFA donors (Donor #1 and Donor #2) with increasing concentrations of a Flag-tagged truncated Tat (Tat$_{86}$) and or a Flag-tagged full length (Tat$_{101}$) Tat expression construct and a β-catenin responsive luciferase reporter

construct. In this experiment we utilized the compound BIO (also known as 6BIO or 6-bromoindirubin-3′-oxime), a potent GSK3β inhibitor, to stimulate β-catenin activity [91]. Although PFAs from two different donors differentially activated β-catenin signaling in the presence of BIO, both $Tat_{86}$ and $Tat_{101}$ effectively suppressed β-catenin activity in a dose-dependent manner (Fig 4A). These data suggest that the effect of Tat on β-catenin activity is not due to Tat function attributed to the second exon.

To further explore the effect of Tat function on β-catenin suppression, we investigated the secretory properties of Tat. Tat is actively secreted to the extracellular space by HIV-1 infected cells; a function which allows Tat to interact with other cells and cell types, potentially modulating cell signaling [45, 46, 92]. The translocation of Tat across membranes relies on a single tryptophan, W11, which is indispensable for Tat's transactivation function [41]. To explore whether the lack of secretory function of Tat affects cellular accumulation and subsequently β-catenin activation, we utilized mutations at Tat W11, Tat W11F and Tat W11Y, in our studies. We co-transfected two PFA donors (Donor #1 and Donor #2) with Flag-tagged full-length Tat expression constructs, $Tat_{101}WT$, $Tat_{101}W11F$, or $Tat_{101}W11Y$ and a β-catenin responsive luciferase reporter construct in the presence of BIO (1mM) to stimulate β-catenin activity. Despite donor-specific differences in the ability to stimulate β-catenin activity, WT Tat was

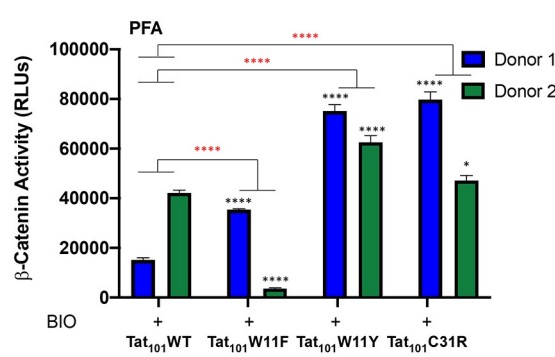

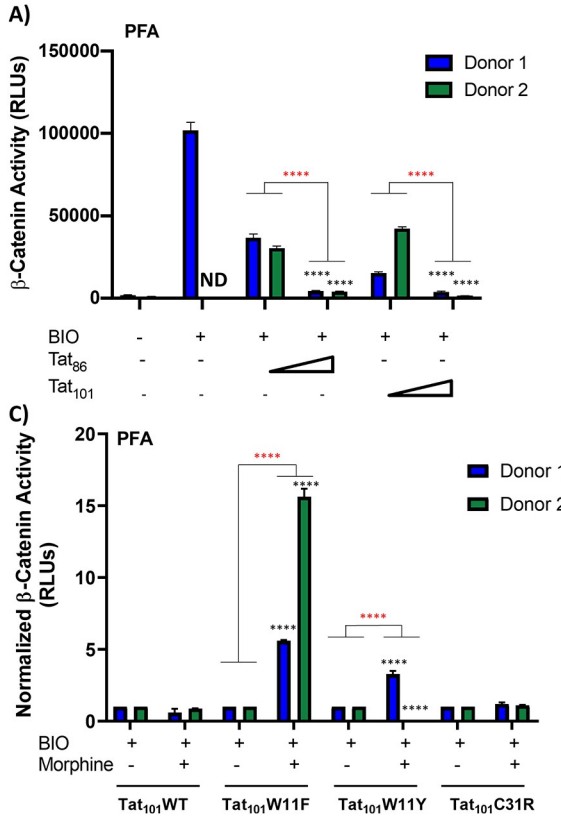

**Fig 4. Tat-mediated suppression of β-catenin activity in PFAs is donor-dependent.** PFAs from Donor 1 and Donor 2 were co-transfected with the TOPFlash β-catenin reporter plasmid and the indicated Flag-Tat expression vectors [A],$Tat_{86}WT$ and $Tat_{101}WT$; 12.5ng, 125ng]; [B,C], $Tat_{101}WT$, $Tat_{101}C31R$, $Tat_{101}W11F$, $Tat_{101}W11Y$; 12.5ng]. 24 hours post transfection, cells were treated with or without BIO (1mM), a β-catenin activator and, where indicated, cells were treated with or without Morphine (500nM). Luciferase activity was measured 24-hours post-treatment. Data shown is the average of technical triplicates from one experiment ± SD. * $p < 0.05$ and **** $p < 0.001$ between low and high Tat or between mutant and WT were calculated by two-way ANOVA with Tukey's multiple comparisons test. Inter-donor comparisons are marked with "*" and intra-donor comparisons are marked with "*".

the most effective at suppressing β-catenin activity compared to the W11F and W11Y Tat mutants (Fig 4B). Residue C31 has been predicted to have a role in HAND due to differences between subtype B and C (C31S) viruses and associated outcomes. In order to examine this in subtype B, Tat Sanger sequences from the CNS AIDS Research and Eradication Study (CARES) Cohort were assessed for any mutations at this position. While C31S was not present, C31R was present and we tested this variant. Similar to W11F and W11Y, we observed no appreciable effects of C31R on Tat-mediated suppression of β-catenin activity (Fig 4B, p = 0.0262).

We next determined the effect of Tat mutants on suppression of β-catenin activity in the presence of morphine. We co-transfected two PFA donors (Donor #1 and Donor #2) with Flag-tagged Tat expression constructs $Tat_{101}WT$, $Tat_{101}W11F$, $Tat_{101}W11Y$, or $Tat_{101}C31R$ and a β-catenin responsive luciferase reporter construct in the presence of BIO (1mM) to stimulate β-catenin activity and in the presence or absence of morphine (500nM). Compared to the untreated control, we observed no change in β-catenin activity with $Tat_{101}C31R$ in either PFA donor (Fig 4C). In contrast, we observed an increase in β-catenin activity in cells expressing $Tat_{101}W11F$ in both PFA donors compared to the untreated control (Fig 4C). Finally, we observed donor-dependent effects in cells expressing $Tat_{101}W11Y$ with Donor 1 supporting an increase, but Donor 2 supporting a suppression of β-catenin activity in the presence of morphine compared to the untreated control (Fig 4C).

## Morphine treatment affects expression of *TrkB*, *NLRP-1*, and *BDNF* genes in astrocytes

Our previous analysis of cellular miRNAs modulated by Tat expression revealed multiple protein targets and pathways that may contribute to neuroinflammation, such as glucocorticoid signaling, axonal guidance, neurotrophin/Trk signaling, and cytokine/chemokine signaling [81]. We were particularly interested in Trk signaling because TrkB is a receptor for BDNF and the dysregulation of this pathway has been implicated in degenerative changes in the CNS [93]. Additionally, *BDNF* gene expression has been shown to be directly regulated by the Wnt/β-catenin pathway [67, 68]. Furthermore, *NLRP-1* was of interest because it is a gene responsive to glucocorticoid signaling; glucocorticoids being inhibitors of the Wnt/β-catenin signaling cascade [94]. Based on these predictions, and the correlates to Wnt/β-catenin signaling, we measured gene expression changes of *TrkB*, *BDNF* and *NLRP-1* in U87MG cells and PFAs using RT-qPCR.

To determine the effect of Tat expression and/or morphine treatment on neuroinflammatory related genes, we transfected astrocytes with the Flag-tagged $Tat_{101}WT$ expression vector in the presence or absence of morphine (500nM). 48 hours post-transfection/treatment, RNA was harvested, processed to cDNA using reverse transcription, and primers were used to amplify target mRNAs (Table 1). Quantitative PCR $C_{(t)}$ values were normalized to *GAPDH* levels and then treatment groups were compared to the control. In U87MG cells, treatment with morphine alone was sufficient to decrease gene expression of *TrkB* and *NLRP-1* (Fig 5A and 5B). Tat expression alone also decreased gene expression of *NLRP-1* but also resulted in a moderate increase in *TrkB* expression (Fig 5A and 5B). Tat and morphine in combination potentiated downregulation of gene expression of both *TrkB* and *NLRP-1* to a greater degree than morphine treatment alone (Fig 5A and 5B). We were not able to detect *BDNF* in U87MG cells (data not shown).

We performed the same experiment in PFAs from three independent donors; in general, we observe similar trends compared to the U87MG experiment, however we do observe donor-to-donor variability. We measured a decrease in *TrkB* gene expression levels in the

presence of morphine and in combination with Tat expression in Donor #1, #2 and #3, (-0.12, -0.63, -0.69, respectively) similar to our data in U87MG cells (Fig 5C). *TrkB* mRNA expression did not appear to be affected by Tat expression alone with fold changes of -0.04, -0.25, 0.1 for Donor #1, #2, and #3, respectively. We also observed morphine-mediated suppression of *NLRP-1* and *BDNF* gene expression in the majority of PFA donors (Fig 5D and 5E). In Donor #2 and Donor #3, Tat expression resulted in a decrease in *NLRP-1* and *BDNF* gene expression (Fig 5D and 5E). Strikingly, Tat and morphine in combination resulted in an increase in *NLRP-1* and *BDNF* gene expression in Donor #2 only, whereas the other two donors under the same conditions resulted in a decrease in gene expression (Fig 5D and 5E). Taken together, morphine treatment results in a consistent loss of expression from neuroinflammatory related genes in U87MG cells and in three PFA donors. Tat expression in morphine treated cells may potentiate this loss of expression, but the phenotype is donor-dependent.

## Tat mutants do not affect the expression of neuroinflammatory related genes in PFAs

Finally, we examined the effect of Tat mutants on the expression of *TrkB*, *NLRP-1* and *BDNF*. We transfected PFAs from three different donors with the indicated Flag-tagged $Tat_{101}$ expression vectors, $Tat_{101}W11F$, $Tat_{101}W11Y$, or $Tat_{101}C31R$. 48 hours post-transfection, RNA was

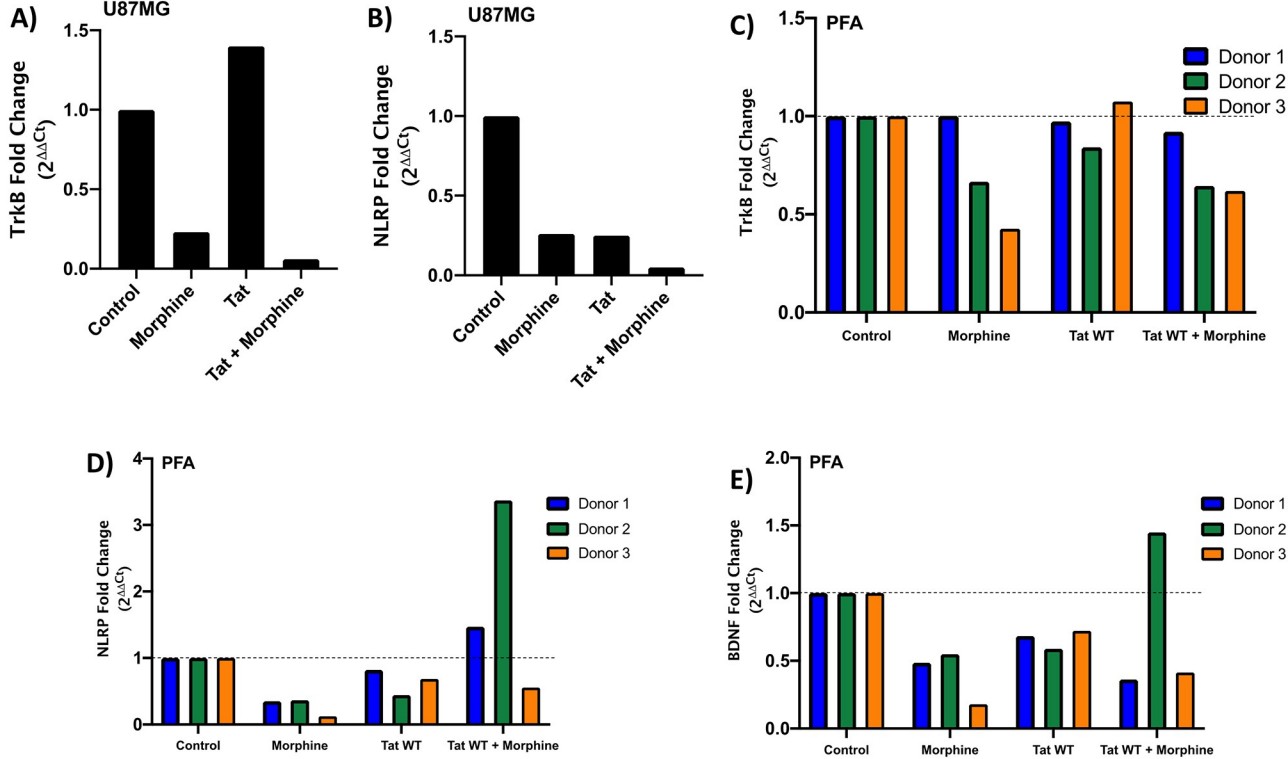

**Fig 5. Treatment of astrocytes with morphine decreases expression of neuroinflammatory genes.** U87MG astrocytes (A and B) and PFAs (C, D and E) from three donors were transfected with either mock plasmid (pUC19) or the Flag-$Tat_{101}$WT expression vector. 24 hours post-transfection, cells were treated with or without Morphine (500nM). 48 hours post-treatment, RNA was harvested and RT-qPCR was performed using primers specific for *TrkB* (A and C), *NLRP-1* (B and D), and *BDNF* (E) mRNA. Fold change values were calculated by normalizing relative mRNA gene expression ($C_t$) to GAPDH ($\Delta C_t$) relative to the control ($\Delta\Delta C_t$). Data is plotted as the fold change of the gene over the conditions ($2^{\Delta\Delta C_t}$). Data shown is from one experiment performed in parallel with three individual PFA donors. For PFAs, significance between conditions and control were calculated by two-way ANOVA with Tukey's multiple comparisons test.

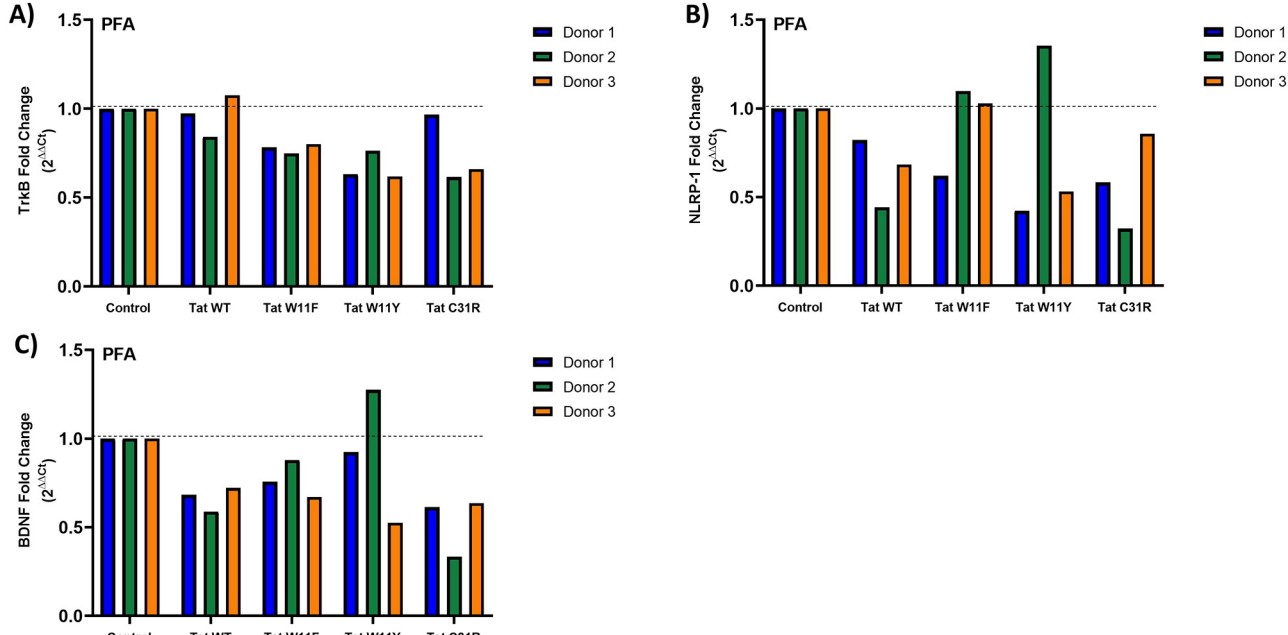

**Fig 6. Tat mutants have variable, donor-dependent effects on expression of neuroinflammatory genes in PFAs.** PFAs from three donors were transfected with the indicated Tat expression vectors, Flag-Tat$_{101}$WT, Flag-Tat$_{101}$C31R, Flag-Tat$_{101}$W11F, or Flag-Tat$_{101}$W11Y. 48 hours post-transfection, RNA was harvested and RT-qPCR was performed using primers specific for *TrkB* (A), *NLRP-1* (B), *and BDNF* (C) mRNA. Fold change values were calculated by normalizing relative mRNA gene expression ($C_t$) to GAPDH ($\Delta C_t$) relative to the control ($\Delta\Delta C_t$). Data is plotted as the fold change of the gene over the conditions ($2^{\Delta\Delta Ct}$). Data shown is from one experiment performed in parallel with three individual PFA donors. Significance between mutant and WT were calculated by two-way ANOVA with Tukey's multiple comparisons test.

harvested and expression levels were determined as in Fig 5. Regardless of the mutant, Tat expression did not significantly alter *TrkB* expression in any of the three PFA donors (Fig 6A). *NLRP*-1 and *BDNF* expression was variably affected by Tat mutant expression in a donor-dependent manner (Fig 6B and 6C). It is therefore difficult to conclude that gain or loss of gene expression can be attributed to specific function of these mutants.

## Discussion

Current estimates of opiate use disorder diagnoses in the United States number near two million, suggesting a significant reciprocal risk between individuals that abuse drugs intravenously and HIV-1 infection rates [30]. Neurocognitive deficits associated with long-term HIV-1 infection can be exacerbated with opiate abuse due to detrimental effects on immune cells. In this study, we explored possible mechanisms underlying the worsening neurocognitive outcomes associated with HIV-1 infection and opiate use. As a result of combined HIV-1 Tat expression and morphine exposure, we observed suppression of β-catenin signaling, a well-characterized neuroprotective pathway against HIV-1 directed neuroinflammation, in human astrocytes.

The dysregulation of β-catenin signaling has been correlated with several neurodegenerative disorders such as Alzheimer's and bipolar disorder [65]. Here, we explored the contribution of opiate exposure, specifically morphine, to HIV-1 Tat mediated suppression of β-catenin signaling. Our study design was strengthened by existing data showing that methamphetamine exposure and Tat expression can downregulate β-catenin signaling in astrocytes [95]. Our data show that morphine can independently suppress β-catenin signaling in

astrocytes in both the U87MG cell line and in primary human cells (Fig 2). This represents a potential mechanism by which opiate abuse may render the CNS more vulnerable to infection and establishment of virus in the CNS.

To further explore the mechanism of Tat suppression of β-catenin activity, we utilized Tat expression constructs that encoded Tat proteins of various lengths or containing functional mutations. In the absence of Tat, a transcriptional inhibitor complex including β-catenin can bind upstream of the viral promoter, preventing the binding of RNA pol II, therefore effectively stalling transcription [60, 61, 64]. In the presence of Tat, Tat antagonizes β-catenin, displaces the complex, and promotes HIV-1 transcription through recruitment of cellular factors. Tat also enhances the degradation of β-catenin. The ability of Tat to suppress β-catenin signaling has been associated with the first 72 amino acids of the Tat protein [63]. In accordance with these findings, we show that both truncated ($Tat_{86}$) and full-length Tat ($Tat_{101}$) can suppress β-catenin signaling (Fig 4A). Tat also contains several functional domains with variable and conserved residues. In particular, a highly conserved tryptophan at residue 11 in the proline-rich domain is required for efficient secretion of Tat. We observed that mutations at this position, W11F and W11Y, which abrogate Tat secretion and uptake, are unable to suppress β-catenin activity compared to WT Tat in the presence or absence of morphine (Fig 4B and 4C). Tat produced during the viral life cycle can be efficiently exported from the infected cell; therefore, this mechanism likely contributes to antigenicity and neurotoxicity in the CNS environment. Loss of suppression by the W11Y mutant suggests that efficient Tat secretion may be important in suppression of β-catenin signaling, perhaps via bystander effect. Therefore, we expect in an environment where few astrocytes are infected that Tat will not induce efficient β-catenin suppression in bystander cells if its secretion function is lost. Transfection of $Tat_{101}$ W11Y at high levels suppressed β-catenin similar to WT (data not shown). We speculate that at low concentrations of Tat, the protein can freely move to a neighboring cell and cause β-catenin inhibition in bystander cells allowing our assay to observe a greater overall reduction in β-catenin activity. Whereas Tat W11Y loses the ability to exit the cell and is limited to downregulating β-catenin in the local astrocyte. Further experiments will be required to test this hypothesis.

HIV-1 Tat also contains a cysteine-rich domain containing highly conserved cysteine residues that are critical for disulfide-bond formation (amino acids 22–37). We also observed a loss of suppression of β-catenin activity by a C31R Tat mutant in the presence or absence of morphine (Fig 4B and 4C). Interestingly, HIV-1 subtype C virus contains a polymorphism at position 31 (C31S) which does not suppress β-catenin activity. This data supports the importance of Tat directed β-catenin suppression in subtype B induced neuroinflammation.

Finally, we investigated the effect of Tat and morphine on genes involved with inflammation and neurodysregulation, *TrkB*, *NLRP-1*, and *BDNF*. We consistently measured a decrease in expression of all genes in the presence of morphine with U87MG cells and PFAs across all donors (Fig 5). We also observed variability in gene expression levels between U87MGs and PFAs (comparing Fig 5A, 5B to 5C–5E) in response to Tat and morphine. These data may highlight differences between the responses of astrocytes depending on developmental stage. Adult exposure to HIV-1 and morphine is likely to have different effects than early life infection or even *in utero* exposure to opiates. Noting gene expression differences between adult astrocytes and fetal astrocytes, this may warrant different approach towards diagnostics, treatment, and management depending on the stage of development of the CNS.

We observed appreciable donor to donor variability in our experiments involving PFAs. Of interest is the reversal of β-catenin activity suppression of all three Tat mutants in the presence of morphine in PFA Donor #1 (Fig 4C). We also observed fluctuation between donors when determining the effect of Tat mutants on gene expression of *TrkB*, *NLRP-1*, and *BDNF* (Fig 6).

It is possible that there are no significant effects of Tat expression on these genes and instead Tat acts at the protein level.

Interestingly, all three genes investigated in this study are regulated by different transcriptional pathways (from literature based on neuronal cells): *BDNF* is directly regulated by the Wnt/ β-catenin pathway, *TrkB* is regulated by PKA/CREB, and *NLRP-1* is controlled by NF-κB [67, 96–98]. These data suggest that although we observe a measurable loss of β-catenin activity with morphine exposure, the presence of this drug is likely affecting other regulatory pathways independent of the β-catenin mechanism. Dysregulation of BDNF has been observed in HIV-1-infected individuals and BDNF levels can be altered by drugs of abuse, including alcohol, heroin, and cocaine [99–102]. These studies are complicated by changing receptor levels and the ratio of mature to pro BDNF in the setting of inflammation [103–105]. Studies suggest that the mu-opioid receptor (MOR) plays a central role in mediating neurotoxic effects. Our results confirm this effect in an *ex vivo* setting with a single cell type. The results from this investigation suggest that abuse of morphine in HIV-1-infected individuals interacts with Tat leading to alterations not only in β-catenin levels, but also its activity in primary astrocyte. This in turn contributes to the underlying mechanism of HAND, furthering our understanding of this poorly understood mechanism.

## Supporting information

**S1 Fig. Expression levels of HIV-1 Tat101.** U87MG cells were transfected with an HIV-1 LTR-Luciferase reporter (pHIV-1-LTR-GL3; 1μg) in the presence or absence of increasing concentrations of a HIV-1 Tat expression vector (pCMV-Tat101-Flag; 8.4ng, 84ng, 840ng) corresponding to the low, medium and high conditions used. pUC19 transfection was included as a control. 48hrs post-transfection, cells were lysed and probed for HIV-1 Tat and β-Actin expression.
(TIF)

## Author Contributions

**Conceptualization:** Kenneth Chen, Angel Lin, Luca Sardo, Michael R. Nonnemacher, Zachary Klase.

**Data curation:** Kenneth Chen.

**Formal analysis:** Kenneth Chen.

**Funding acquisition:** Zachary Klase.

**Investigation:** Kenneth Chen, Thienlong Phan, Angel Lin, Luca Sardo.

**Methodology:** Thienlong Phan, Anthony R. Mele.

**Resources:** Anthony R. Mele.

**Supervision:** Zachary Klase.

**Writing – original draft:** Kenneth Chen.

**Writing – review & editing:** Michael R. Nonnemacher, Zachary Klase.

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
