## [Decision Letter · Decision Letter 0]

2 Jul 2019

PONE-D-19-16014

Morphine exposure exacerbates HIV-1 Tat driven changes to neuroinflammatory factors in cultured astrocytes

PLOS ONE

Dear Dr Klase,

Thank you for submitting your manuscript to PLOS ONE. After careful consideration, we feel that it has merit but does not fully meet PLOS ONE’s publication criteria as it currently stands. Therefore, we invite you to submit a revised version of the manuscript that addresses the points raised during the review process.

We would appreciate receiving your revised manuscript by Aug 16 2019 11:59PM. To enhance the reproducibility of your results, we recommend that if applicable you deposit your laboratory protocols in protocols.io, where a protocol can be assigned its own identifier (DOI) such that it can be cited independently in the future. For instructions see: http://journals.plos.org/plosone/s/submission-guidelines#loc-laboratory-protocols

We look forward to receiving your revised manuscript.

Kind regards,

Navneet K. Dhillon

Academic Editor

PLOS ONE

Journal Requirements:

Please provide an amended Funding Statement that declares *all* the funding or sources of support received during this specific study (whether external or internal to your organization) as detailed online in our guide for authors at http://journals.plos.org/plosone/s/submit-now.  

Please state what role the funders took in the study.  If any authors received a salary from any of your funders, please state which authors and which funder. If the funders had no role, please state: "The funders had no role in study design, data collection and analysis, decision to publish, or preparation of the manuscript."

3. We note you have included a table to which you do not refer in the text of your manuscript. Please ensure that you refer to Table 1 in your text; if accepted, production will need this reference to link the reader to the Table.

Reviewers' comments:

Reviewer's Responses to Questions

**Comments to the Author**

1. Is the manuscript technically sound, and do the data support the conclusions?

Reviewer #1: Yes

Reviewer #2: Partly

2. Has the statistical analysis been performed appropriately and rigorously? 

Reviewer #1: No

Reviewer #2: No

3. Have the authors made all data underlying the findings in their manuscript fully available?

Reviewer #1: Yes

Reviewer #2: Yes

4. Is the manuscript presented in an intelligible fashion and written in standard English?

Reviewer #1: Yes

Reviewer #2: Yes

5. Review Comments to the Author

Reviewer #1: The manuscript by Chen and colleagues is reporting the effects of morphine on the suppression of b-catenin in the presence of HIV Tat. This is a relevant manuscript, with important findings. Yet, the manuscript presentation has several problems that need to be addressed, including focus and flow. They have been itemized below, not in order of relevance but as they appeared in the text, followed by a general summary. We believe that the findings are important, but major revision needs to be performed.

1. The abstract needs to be revised for missing verbs and for improving clarity of goals, methods and results. For instance, the study is performed in astrocytes, cell lines and primary cultures. However, the relevance of studying the effects of Tat on astrocytes is not clear, in the abstract. The results should be better outlined. It is right now too brief, and short on explaining the actual findings. The importance of studying the effects on b-catenin is also not clear in the abstract. The study of miRNA or targets is not stated, or why.

2) The Introduction has a lot on HAND and Tat (missing citations and important literature), but it is short on offering background about what was the actual study, which is a mechanism, not the disorder. The model, the outcomes, and their relevance all should be stated upfront.

3) miRNA in the Introduction is disconnected (line 71). Terrible flow.

4) Literature on the effects of Tat on RNA PolII and on dopamine pathway is outdated.

5) Case for studying astrocytes, rather than other "physiologically relevant" cells, in the context of HIV, and in particular, neurological disorders, is not sufficiently strong.

6) The explanations about b-catenin (line 94) may have to be focused towards astrocytes.

7) Citation on numbers of opioids and HIV, and on inflammatory molecules (which by the way are not the ones investigated in the study) are needed (line 105 and 110).

8) The significance of Tat mutants is not stated in Introduction.

9) Results chapter start with miRNA, but I don't see results on miRNAs. Unfocused.

10) Overall, It is not shown whether the transfection with low, moderate or high plasmid really corresponds to low, moderate and high protein expression, upon transduction. This is desirable, to demonstrate the efficiency of the transfection and the protein levels (which are rather more important).

11) The three different models (cell line, primary astrocytes, and donors) are not well separated or justified, causing confusion and problems of flow.

12) It is not clear why in primary astrocytes there are only two doses with a 2log difference.

13) Tat only controls should be included in all graphs. It is very hard to appreciate the effects of interaction if not ALL controls are included in the graphs.

14) Reference for serum concentration of morphine found in subjects who overdosed on heroin is not provided (line 144).

15) IN line 180, all of a sudden there are donors. Human, mice, monkeys, fetal, purity of cultures, and mainly why? Only two, and with opposite results? This doesn't seem to add anything. There is zero power in this experiment, even with a paired analysis. There are two choices here. Drop the donor experiments and make a better case for astrocytes in general, introduce and discuss better the effects on b-catenin, or increase the number of donors, with data on sex, age, pre-exposure, HIV status, everything that is missing here, including statistics, to make sense. Another strong alternative is to show that the effects are reverted with interventions that restore b-catenin levels.

16) Line 190, different mutants give different results. Nowhere we see why is this relevant, whether they differ in structural properties, and implications. All this should be in the discussion.

17) Line 205. All of a sudden we have NLRP-1 and other things that were not introduced at all. Again 3 donors. AND, there no discussion about individual variability at all.

18) IN the discussion also, comments on CCL5, iNOS, genes that were not measured.

19) Paired Student's t test is not correct. Student's t test is unpaired.

Very hard to follow. Rationale has to be better developed. Focus has to be largely improved. Further experiments to increase the power of readouts, with additional donors, including acknowledgment of factors that may affect individual responses in this pathway, as well as interventional studies, are recommended.

Reviewer #2: The manuscript entitled "Morphine exposure exacerbates HIV-1 Tat driven changes to neuroinflammatory

factors in cultured astrocytes "

Since HAND increases in drug abusers, this study was directed to elucidate the role morphine on the effect of Tat HIV-1 and Tat mutants associated with neurocognitive impairment mediated by the β-catenin suppression and inflammatory processes.

Although the data may be interesting, the manuscript has serious errors that must be resolved before it is acceptable for publication.

2.- Some paragraphs do not have a bibliographic support and there are multiple typographical, orthographic and verb conjugation errors.

3.- Throughout the writing different styles are noticed, with some secctions well written and others are not.

4.- Major changes have to be made in the methods section.

4.1. The subtitles in general are inadequate

4.2. Given that the primary cultures of astrocytes are of fetal origin, it is important to inform about the time during the fetal development in which the cells were taken, how many donors are included in the study and if the experiments were performed in same donors?, please mention culture conditions for primary cultures and the cell line (was the same medium was used for both’), mention the passage of the culture in which the studies were made.

4.3. The authors must demonstrate that the primary cultures are really astrocytes and not neural stem cells, which are also positive to GFAP. Immunohistochemistry is recommended for neural stem markers.

4.3. Furthermore, depending on the fetal age in which the cells are obtained, they have different potential for differentiation. The fact that they are proliferative indicates that they are stem cells since the differentiated astrocytes do not proliferate, unless they are cancerous.

4.4. The luciferase assay is not in the methods

4.5. Please include the data of all suppliers and the information on the antibody ID used (http://antibodyregistry.org/).

4.6. It is not clear how long after the transfections and treatments the tests were performed.

4.7. In the section of reverse transcription authors mention that it was done as for the microRNAs, but in the manuscript, there is no analysis of these molecules. Please describe how the reverse transcriptions were made.

4.8. The use of 1μl of cDNA without quantifying the amount of cDNA used in the PCR reactions is inadequate, please mention the amount of cDNA used.

4.9. Graphs on PCRs are inaccurate some are as relative expression (if so, the control group should have a value of 1), and others as xmRBNA/GAPDH. The authors are asked to clarify which method of expression analysis is used, and they are recommended to use 2-ΔΔCT.

4.9. Statistic section is unclear, why did the authors used Paired Student’s t test. And why after ANOVA a pairwise analysis was performed and not a multiple comparison test.

Results:

1. The images are of very poor quality.

2. In some results they refer to the donor number and in others they do not.

3. The luciferase assay are confusing, perhaps because it is not described in the methods.

4. Not all figure legends have the statistical test used.

5. In cases where ANOVA was used, which post-test was performed.

6. It is inappropriate to use the Student's t to compare more than two groups (example Figure 1)

7. The number of experiments is not clear.

The discussion does not have a clear focus and repeats a large part of the results, which makes it tedious and very long.

6. PLOS authors have the option to publish the peer review history of their article (what does this mean?). If published, this will include your full peer review and any attached files.

Reviewer #1: Yes: Maria Cecilia Garibaldi Marcondes

Reviewer #2: No

---

## [Author Response · Author response to Decision Letter 0]

13 Jan 2020

A fully formatted and color coded version of this response to your comments exists at the end of the combined PDF (at least in our view). This version may be easier to read than the comments copied below. 

We wish to thank the reviewers for their time and comments. The critiques provided were all very helpful and we have addressed them individually below. It should be noted that we wanted to do much more. In setting out to do these revisions we originally planned to expand our use of primary astrocytes so that all figures would include data from three to five different donors. This would have increased our statistical power, allowed comparison between immortalized astrocytes (U87MG) and primary cells, and have provided additional controls for each experiment. However, recent changes in the NIH guidelines on the use of fetal material has made it impossible to obtain new cultures. We had some material on hand and were able to add data points to some of the existing analyses, but not as much as we would have liked. The text of the manuscript has been updated extensively. We believe this new text provides a much clearer explanation of the rationale for the experiments and a cleaner discussion of the findings.

Specific points are addressed below:

Reviewer #1: The manuscript by Chen and colleagues is reporting the effects of morphine on the suppression of b-catenin in the presence of HIV Tat. This is a relevant manuscript, with important findings. Yet, the manuscript presentation has several problems that need to be addressed, including focus and flow. They have been itemized below, not in order of relevance but as they appeared in the text, followed by a general summary. We believe that the findings are important, but major revision needs to be performed.

1. The abstract needs to be revised for missing verbs and for improving clarity of goals, methods and results. For instance, the study is performed in astrocytes, cell lines and primary cultures. However, the relevance of studying the effects of Tat on astrocytes is not clear, in the abstract. The results should be better outlined. It is right now too brief, and short on explaining the actual findings. The importance of studying the effects on b-catenin is also not clear in the abstract. The study of miRNA or targets is not stated, or why.

We agree with the reviewer that the abstract would benefit from revision. We have rewritten the abstract as follows:

“Despite antiretroviral therapy (ART), human immunodeficiency virus type-1 (HIV-1) infection results in neuroinflammation of the central nervous system (CNS) that can cause HIV-associated neurocognitive disorders (HAND). The molecular mechanisms involved in the development of HAND are unclear, however, they are likely due to both direct and indirect consequences of HIV-1 infection and inflammation of the CNS. Additionally, opioid abuse in infected individuals has the potential to exacerbate HIV-comorbidities, such as HAND. Although restricted for productive HIV replication, astrocytes (comprising 40-70% of all brain cells) likely play a significant role in neuropathogenesis in infected individuals due to the production and response of viral proteins. The HIV-1 protein Tat is critical for viral transcription, causes neuroinflammation, and can be secreted from infected cells to affect uninfected bystander cells. The Wnt/β-catenin signaling cascade plays an integral role in restricting HIV-1 infection in part by negatively regulating HIV-1 Tat function. Conversely, Tat can overcome this negative regulation and inhibit β-catenin signaling by sequestering the critical transcription factor TCF-4 from binding to β-catenin. Here, we aimed to explore how opiate exposure affects Tat-mediated suppression of β-catenin in astrocytes and the downstream modulation of neuroinflammatory genes. We observed that morphine can potentiate Tat suppression of β-catenin activity in human astrocytes. In contrast, Tat mutants deficient in secretion, and lacking neurotoxic effects, do not affect β-catenin activity in the presence or absence of morphine. Finally, morphine treatment of astrocytes was sufficient to reduce the expression of genes involved in neuroinflammation. Examining the molecular mechanisms of how HIV-1 infection and opiate exposure exacerbate neuroinflammation may help us inform or predict disease progression prior to HAND development.”

2) The Introduction has a lot on HAND and Tat (missing citations and important literature), but it is short on offering background about what was the actual study, which is a mechanism, not the disorder. The model, the outcomes, and their relevance all should be stated upfront.

 We agree with the reviewer that the Introduction was misfocused. We have now re-written the entire Introduction to hopefully improve the clarity of the study.

3) miRNA in the Introduction is disconnected (line 71). Terrible flow.

We agree with the reviewer that introducing the contribution of Tat to the dysregulation of cellular miRNA pathways is out of context at this point in the introduction. We have removed this statement.

4) Literature on the effects of Tat on RNA PolII and on dopamine pathway is outdated.

We thank the reviewer for bringing this to our attention. We have now included two additional citations to recent reviews on Tat function. We have also removed the statement on Tat effects on the dopamine pathway as it is not relevant to the current study.

5) Case for studying astrocytes, rather than other "physiologically relevant" cells, in the context of HIV, and in particular, neurological disorders, is not sufficiently strong.

We thank the reviewer for bringing this to our attention. We have largely rewritten the Introduction to provide a stronger case for studying HIV-1 and neurological disorders in astrocytes as opposed to other cells of the CNS. Some newly added statements include:

“CNS-resident, long-lived cell types such as monocytes, macrophages, and astrocytes have all been implicated in HIV-1 infection and neuroinflammation, though the establishment of latent and prolonged infection in these cell types within the CNS remains controversial [15]. Astrocytes, specifically, can be permissive to HIV-1 infection, but these cells are restrictive to productive replication, and there is limited data demonstrating infection in vivo [16-18]. Within the CNS, astrocytes maintain the homeostatic balance and play key roles in mediating cerebral blood flow by maintaining the BBB, providing neuronal support through neurotransmitter processing, regulating nutrients and growth factors, and modulating inflammation [19-22] . HIV-1 does not infect neurons, implying that development of HAND might be mediated indirectly, for example through astrocyte dysregulation and inflammation. A measurable percentage of astrocytes in the CNS can harbor HIV-1 integrated DNA, making astrocytes a likely contributor in the development of HAND [23-26].”

6) The explanations about b-catenin (line 94) may have to be focused towards astrocytes.

 We agree with the reviewer and have now provided information regarding the role of B-catenin in astrocytes:

“The Wnt/ β-catenin pathway has been implicated in suppressing replication of HIV-1 in multiple cell types, including astrocytes, and conversely, downregulation of Wnt/β-catenin signaling by inflammatory cytokines promotes HIV-1 replication [59-63]. Interestingly, HIV-1 Tat has also been shown to inhibit Wnt/β-catenin signaling in astrocytes, therefore alleviating, in part, the negative regulation of Tat-mediated transactivation of the LTR by β-catenin [60, 63, 64].” 

… 

“Compared to other cell types and despite robust levels of signaling, less is known about the role of Wnt/ β-catenin in astrocyte function [59, 62, 65]. Approximately 150 β-catenin-regulated genes are estimated in primary astrocytes are comprising at least five broad categories including inflammation, transport, exocytosis, apoptosis, and trafficking [61]. Of these, β-catenin responsive genes that are neuroinflammatory in nature include BDNF (Brain-Derived Neurotrophic Factor), a member of the BDNF/TrkB (Tropomyosin receptor kinase B) signaling axis [66-68].”

7) Citation on numbers of opioids and HIV, and on inflammatory molecules (which by the way are not the ones investigated in the study) are needed (line 105 and 110).

We agree with the reviewer that a citation on the numbers of opioids and HIV is needed. We have revised the statement for accuracy as follows:

“In addition to neurotoxicity associated with HIV-1 infection, mounting evidence suggests exacerbation of these effects in the presence of opiates [69-72]. Illicit use and drug abuse contribute significantly to HIV-1 infections and transmission [73], therefore elucidating the interaction between the cells of the CNS, opiates, and HIV-1 Tat is necessary to understanding HAND disease progression.” 

We have also removed the information about inflammatory molecules as they are not related to the study.

8) The significance of Tat mutants is not stated in Introduction.

We agree with the reviewer that stating the significance of the Tat mutants in the Introduction will add to the clarity of the manuscript. We have now added the following statements to the Introduction: 

“HIV-1 is subject to genetic variation, which may alter the functionality of the encoded viral proteins. In vitro, Tat expressing cells have been demonstrated to readily secrete Tat extracellularly, a function mediated by a tryptophan at amino acid position 11 (W11), as well as residues 49-51 within the basic domain [41, 45-47]. Tat variants containing mutations at this position, such as W11F or W11Y, are deficient in both secretion from Tat-producing cells and cytoplasmic translocation in recipient cells [46, 48]. In the CNS, Tat can also be secreted from HIV-1 infected cells, such as macrophages and astrocytes, which results in direct neurotoxicity; therefore, Tat secretion-deficient mutants may alleviate this phenotype within patients [49-55]. Finally, mutations at a conserved cysteine at amino acid position 31, C31, which is within the cysteine-rich domain, have been shown to abrogate synaptodendritic injury in neurons and normalize levels of amyloid beta production, therefore conferring protection from Tat-mediated neurotoxicity [41, 56-58].”

9) Results chapter start with miRNA, but I don't see results on miRNAs. Unfocused.

We agree with the reviewer that the Results section required additional clarification as to our interest in pursuing this study. We have modified the beginning of the Results section as follows: 

“We have recently determined that HIV-1 Tat-mediated suppression of specific miRNAs affects downstream signaling from the Wnt/�-catenin pathway [78]. The modulation of �-catenin activity by Tat could alleviate repression of basal HIV-1 transcription, resulting in enhanced neuroinflammation in the CNS [60, 63, 64]. To confirm our previous results and to address the ability of Tat to modulate β-catenin activity in physiologically relevant cells, we utilized the U87MG astrocytic cell line and primary fetal astrocyte (PFA) donors in our study.”

10) Overall, It is not shown whether the transfection with low, moderate or high plasmid really corresponds to low, moderate and high protein expression, upon transduction. This is desirable, to demonstrate the efficiency of the transfection and the protein levels (which are rather more important).

We agree and have added a new supplemental figure demonstrating detection of Tat expression in the three conditions. Tat is clearly visible in the moderate and high doses, but not in the low. However, an LTR-Luciferase assay reveals strong activation (5-fold) at the lowest dose, supporting the presence of Tat protein.

11) The three different models (cell line, primary astrocytes, and donors) are not well separated or justified, causing confusion and problems of flow.

We agree with the reviewer that clarification of which cells were used for given experiments was needed. We have now indicated use of either the U87MG astrocytic cell line or any of three primary fetal astrocyte (PFA) donors (Donor 1, Donor 2, Donor 3) on each figure. We have also clarified which cells were used in each relevant section of the Results and in the Figure Legends. Finally, the source of the cells has been clarified in the Materials and Methods.

As noted above, recent changes in the NIH guidelines on the use of fetal material has made it impossible to obtain the new cultures needed to expand on the data as originally presented. Using material on hand we were able to include new data in some of the figures.

12) It is not clear why in primary astrocytes there are only two doses with a 2log difference.

We agree with the reviewer that some clarification is needed for the details of the LiCl treatment, Morphine treatment, and Tat transfection between U87MG cells and PFAs. The LiCl concentrated used (50mM) is consistent across all experiments. In U87MG cells, we include additional concentrations of Morphine and Tat due to the ease and adaptability of using an immortalized cell line. Based on the results from the U87MG-based experiments, we optimized conditions for the PFA-based experiments, which are more challenging and of limited supply. For example, we see little to no difference in the effect of 0.35 Tat on B-catenin activity (Compared to the untransfected control and compared to 3.5ng Tat; Figure 1A and Figure 2A), therefore, this concentration was not included in the experiment in Figure 1B or Figure 2B. Additionally, we expected to see the largest effect on B-catenin activity in PFAs at the highest concentration of Morphine tested (500nM).

We have indicated the above information in revised versions of the Figures and Figure Legends which we hope will clarify how the experiments were performed.

13) Tat only controls should be included in all graphs. It is very hard to appreciate the effects of interaction if not ALL controls are included in the graphs.

 The figures now include Tat only controls. When using a B-catenin activator (LiCl or Bio) we have not included analyses of Tat effect on the cells in the absence of activator. In these systems we have needed to activate B-catenin in order to detect activity and see changes. Figure 4C does not include no Tat control as the figure exists to compare within the different mutants tested – in this figure Tat WT serves as the control. Additionally, we now include a Western blot demonstrating the varying amounts of Tat detectable at the amounts transfected.

14) Reference for serum concentration of morphine found in subjects who overdosed on heroin is not provided (line 144).

The statement describing the rationale for the amount of morphine use has been adjusted and no includes proper references.

15) IN line 180, all of a sudden there are donors. Human, mice, monkeys, fetal, purity of cultures, and mainly why? Only two, and with opposite results? This doesn't seem to add anything. There is zero power in this experiment, even with a paired analysis. There are two choices here. Drop the donor experiments and make a better case for astrocytes in general, introduce and discuss better the effects on b-catenin, or increase the number of donors, with data on sex, age, pre-exposure, HIV status, everything that is missing here, including statistics, to make sense. Another strong alternative is to show that the effects are reverted with interventions that restore b-catenin levels.

We agree with the reviewer that additional clarification was needed at this point in the manuscript. We believe that steps taken to clarify the difference between the U87MG cell line used and PFA donors (discussed in point #11) will help alleviate the confusion. We agree with the reviewer that two donors may be insufficient to draw significant conclusions. As noted above, the resources needed to perform these experiments (inclusion of astrocytes from additional donors or extensive new experiments in the cultures shown) are not available due to changes in NIH funding guidelines. The error bars present in the figures are the result of biological replicates performed with the same donor astrocytes. As such, we that descriptions of Tat and morphine changes in each donor are valid. The text has been updated to provide clear explanation of where changes are common between the donors used.

All donor material is HIV negative and derives from gestational age of 16-18 weeks as noted in materials and methods.

16) Line 190, different mutants give different results. Nowhere we see why is this relevant, whether they differ in structural properties, and implications. All this should be in the discussion.

We agree with the reviewer that additional clarification is needed for both the two different forms of Tat, Tat86 and Tat101 as well as the Tat mutants C31R, W11Y, and W11F. We have added the following clarifying statements to the appropriate section of the Results: 

“Tat is encoded by HIV-1 on two exons, the first of which is sufficient for the primary function of Tat to act as the viral transactivator and drive the transcriptional elongation of viral genes [83-85]. The functions of Tat encoded by the second exon contribute to viral replication and the modulation of interferon-stimulated genes in antigen presenting cells [86, 87]. Widely used laboratory-adapted strains of HIV-1 produce a truncated form of Tat, encoded by the first exon plus 14 amino acids of exon 2, Tat86. The full-length transcript, however, encodes for the full-length form of Tat, Tat101. To determine if the two different Tat variants have an effect on suppression of β-catenin activity, we co-transfected two PFA donors (Donor #1 and Donor #2) with increasing concentrations of a Flag-tagged truncated Tat (Tat86) and or a Flag-tagged full length (Tat101) Tat expression construct and a β-catenin responsive luciferase reporter construct. In this experiment we utilized the compound BIO (also known as 6BIO or 6-bromoindirubin-3′-oxime), a potent GSK3� inhibitor, to stimulate �-catenin activity [88]. Although PFAs from two different donors differentially activated β-catenin signaling in the presence of BIO, both Tat86 and Tat101 effectively suppressed β-catenin activity in a dose-dependent manner (Figure 4A). These data suggest that the effect of Tat on β-catenin activity is not due to Tat function attributed to the second exon.” 

We have also added the following clarifying statements to the appropriate section of the Discussion:

“To further explore the mechanism of Tat suppression of β-catenin activity, we utilized Tat expression constructs that encoded Tat proteins of various lengths or containing functional mutations. In the absence of Tat, a transcriptional inhibitor complex including β-catenin can bind upstream of the viral promoter, preventing the binding of RNA pol II, therefore effectively stalling transcription [60, 61, 64]. In the presence of Tat, Tat antagonizes β-catenin, displaces the complex, and promotes HIV-1 transcription through recruitment of cellular factors. Tat also enhances the degradation of β-catenin. The ability of Tat to suppress β-catenin signaling has been associated with the first 72 amino acids of the Tat protein [63]. In accordance with these findings, we show that both truncated (Tat86) and full-length Tat (Tat101) can suppress β-catenin signaling (Figure 4A). Tat also contains several functional domains with variable and conserved residues. In particular, a highly conserved tryptophan at residue 11 in the proline-rich domain is required for efficient secretion of Tat. We observed that mutations at this position, W11F and W11Y, which abrogate Tat secretion and uptake, are unable to suppress β-catenin activity compared to WT Tat in the presence or absence of morphine (Figure 4B, C). Tat produced during the viral life cycle can be efficiently exported from the infected cell; therefore, this mechanism likely contributes to antigenicity and neurotoxicity in the CNS environment. Loss of suppression by the W11Y mutant suggests that efficient Tat secretion may be important in suppression of β-catenin signaling, perhaps via bystander effect. Therefore, we expect in an environment where few astrocytes are infected that Tat will not induce efficient β-catenin suppression in bystander cells if its secretion function is lost. Transfection of Tat101 W11Y at high levels suppressed β-catenin similar to WT (data not shown). We speculate that at low concentrations of Tat, the protein can freely move to a neighboring cell and cause β-catenin inhibition in bystander cells allowing our assay to observe a greater overall reduction in β-catenin activity. Whereas Tat W11Y loses the ability to exit the cell and is limited to downregulating β-catenin in the local astrocyte. Further experiments will be required to test this hypothesis. 

HIV-1 Tat also contains a cysteine-rich domain containing highly conserved cysteine residues that are critical for disulfide-bond formation (amino acids 22-37). We also observed a loss of suppression of β-catenin activity by a C31R Tat mutant in the presence or absence of morphine (Figure 4B, C). Interestingly, HIV-1 subtype C virus contains a polymorphism at position 31 (C31S) which does not suppress β-catenin activity. This data supports the importance of Tat directed β-catenin suppression in subtype B induced neuroinflammation.”

17) Line 205. All of a sudden we have NLRP-1 and other things that were not introduced at all. Again 3 donors. AND, there no discussion about individual variability at all.

We agree with the reviewer that additional introduction of the genes of interest in this study is needed. We have now added the following statements to the Introduction:

“Approximately 150 β-catenin-regulated genes are estimated in primary astrocytes are comprising at least five broad categories including inflammation, transport, exocytosis, apoptosis, and trafficking [61]. Of these, β-catenin responsive genes that are neuroinflammatory in nature include BDNF (Brain-Derived Neurotrophic Factor), a member of the BDNF/TrkB (Tropomyosin receptor kinase B) signaling axis [66-68].”

We have also added the following statements to the Results:

“Our previous analysis of cellular miRNAs modulated by Tat expression revealed multiple protein targets and pathways that may contribute to neuroinflammation, such as glucocorticoid signaling, axonal guidance, neurotrophin/Trk signaling, and cytokine/chemokine signaling [78]. We were particularly interested in Trk signaling because TrkB is a receptor for BDNF and the dysregulation of this pathway has been implicated in degenerative changes in the CNS [90]. Additionally, BDNF gene expression has been shown to be directly regulated by the Wnt/β-catenin pathway [67, 68]. Furthermore, NLRP-1 was of interest because it is a gene responsive to glucocorticoid signaling; glucocorticoids being inhibitors of the Wnt/β-catenin signaling cascade [91]. Based on these predictions, and the correlates to Wnt/β-catenin signaling, we measured gene expression changes of TrkB, BDNF and NLRP-1 in U87MG cells and PFAs using RT-qPCR.”

18) IN the discussion also, comments on CCL5, iNOS, genes that were not measured.

We apologize for the confusion. Mention of genes that were not measured in this study have been removed.

19) Paired Student's t test is not correct. Student's t test is unpaired.

We agree with the reviewer and have revised the statistical tests used throughout the manuscript. The tests used have now been indicated in the Materials and Methods and in the Figure Legends. Where indicated that Student’s t test was used, we now used unpaired.

 

Reviewer #2: The manuscript entitled "Morphine exposure exacerbates HIV-1 Tat driven changes to neuroinflammatory factors in cultured astrocytes "

Since HAND increases in drug abusers, this study was directed to elucidate the role morphine on the effect of Tat HIV-1 and Tat mutants associated with neurocognitive impairment mediated by the β-catenin suppression and inflammatory processes.

Although the data may be interesting, the manuscript has serious errors that must be resolved before it is acceptable for publication.

2.- Some paragraphs do not have a bibliographic support and there are multiple typographical, orthographic and verb conjugation errors.

We agree with the reviewer and hope that the largely revised version of the manuscript has addressed the grammatic, syntax, and bibliographic errors.

3.- Throughout the writing different styles are noticed, with some sections well written and others are not.

We agree with the reviewer and to improve clarity and read-ability, the majority of the text has been re-written to maintain “one voice” throughout the manuscript.

4.- Major changes have to be made in the methods section.

We agree with the reviewer and have rewritten the Materials and Methods section to include all pertinent information including the information outlined in the subsections below.

4.1. The subtitles in general are inadequate

We agree with the reviewer and have re-written the subtitles to be more inclusive and descriptive.

4.2. Given that the primary cultures of astrocytes are of fetal origin, it is important to inform about the time during the fetal development in which the cells were taken, how many donors are included in the study and if the experiments were performed in same donors?, please mention culture conditions for primary cultures and the cell line (was the same medium was used for both’), mention the passage of the culture in which the studies were made.

We agree with the reviewer that additional information is needed. We have now included a statement about the origin of the primary cells as follows:

“Fetal brain tissue (gestational age, 16-18 weeks) was obtained from elective abortion procedures performed in full compliance with National Institutes of Health and Temple University ethical guidelines. The tissue was washed with cold Hanks balanced salt solution (HBSS), meninges and blood vessels were removed. For glial cultures, tissue in HBSS was digested with 0.25% trypsin for 30 min at 37C. Trypsin was neutralized with FBS, and the tissue was further dissociated to obtain single-cell suspensions. For glial cultures, cells were plated in mixed glial growth media (same as Astrocyte Growth Media, but with 10% FBS). The mixed culture was maintained under 10% Co2 for 5 days, and the medium was fully replaced to remove any cell debris. To enrich for microglia and astrocytes, flasks were placed on an orbital shaker for 14-18 hours at 200 rpm in growth media. Detached cells constitute the microglial component of the culture and were collected and plated into a new flask containing microglial media. Monolayers that remain after shaking constitute the astrocytes and are fed with astrocyte media.”

We have now included culture conditions for the U87MG cell line and the primary cells (PFAs) as follows:

“U87MG cells were maintained in Eagles Minimal Essential media (EMEM) supplemented with 10% Fetal Bovine Serum (FBS), 4.5g/L of L-glutamine, and 1% penicillin/streptomycin. Primary fetal astrocyte (PFAs) were maintained in Astrocyte Growth Media composed of Dulbecco’s Modified Eagle Media (DMEM) supplemented with 15% FBS (exosome free), 4.5 g/L of L-glutamine, 100 ug/mL gentamicin, 10 ug/mL Amphotericin B, and 20ug/mL insulin.”

4.3. The authors must demonstrate that the primary cultures are really astrocytes and not neural stem cells, which are also positive to GFAP. Immunohistochemistry is recommended for neural stem markers.

We agree with the reviewer and have now included the following statement regarding the purity of the primary cells: 

“Purity of cell type specific cultures was assessed by immunolabeling with anti-GFAP and -GLAST1 for astrocytes, -lba-1 and -CD11b for microglia and -MAP2 or -neurofilament for neurons.”

4.3. Furthermore, depending on the fetal age in which the cells are obtained, they have different potential for differentiation. The fact that they are proliferative indicates that they are stem cells since the differentiated astrocytes do not proliferate, unless they are cancerous.

We agree that the age of the cells, both in terms of when during development they are derived AND when in the lifetime of an individual the cells are examined. In this case, the cells are derived from the fetal tissue by the cell core at Temple University Comprehensive NeuroAIDS Center. They are expanded during this process as they are differentiated appropriately and will expand a limited number of times in culture and therefor have a certain amount of ‘stemness’ The manuscript includes mention of the difference in age of the cultures used.

4.4. The luciferase assay is not in the methods

We thank the reviewer for bringing this to our attention. We have now included the B-catenin luciferase assay in the Materials and Methods section as follows: 

“24 hours post-transfection or post-treatment, where indicated, cells were lysed with passive lysis buffer (Promega). Cell lysates were incubated in equal volumes with the DualGlo luciferase reagent (Promega) and luciferase was measured in a luminometer. Data shown represent the average of technical triplicates.”

4.5. Please include the data of all suppliers and the information on the antibody ID used (http://antibodyregistry.org/).

We thank the reviewer for the comment. We have now added all relevant antibody information into the Materials and Methods as follows:

“The �-total β-catenin antibody (ThermoFisher, cat #MA1-301) and the �-active β-catenin antibody (US Biological Life Sciences, cat# C2069-47) were used at a 1:1000 dilution. The antibody against total β-catenin recognizes all forms of the protein, however, the antibody against active β-catenin recognizes unphosphorylated Ser32 and Ser37. The �-GAPDH antibody (Abcam, cat# ab9485) was used at a 1:3000 dilution.”

4.6. It is not clear how long after the transfections and treatments the tests were performed.

We thank the reviewer for bringing this to our attention. We have now included additional clarifying information on the experimental design both in the Materials and Methods and in the Figure Legends. In general, cells were transfected the day after plating, when cells were approximately 90% confluent. 24 hours post-transfection, at approximately 90% transfection efficiency, cells were treated with the indicated compounds. 24 hours post-treatment, cells were collected and lysed either for luciferase assay or Western blot. 48-hours post-treatment, cells were collected for qPCR analysis.

4.7. In the section of reverse transcription authors mention that it was done as for the microRNAs, but in the manuscript, there is no analysis of these molecules. Please describe how the reverse transcriptions were made.

4.8. The use of 1μl of cDNA without quantifying the amount of cDNA used in the PCR reactions is inadequate, please mention the amount of cDNA used.

We have removed the erroneous text and include proper methods on RT for the qPCR

4.9. Graphs on PCRs are inaccurate some are as relative expression (if so, the control group should have a value of 1), and others as xmRBNA/GAPDH. The authors are asked to clarify which method of expression analysis is used, and they are recommended to use 2-ΔΔCT.

We thank the reviewer for bringing this to our attention. We have now re-plotted the data in Figure 5 as “Gene Fold change 2^ ΔΔCT.” Values less than one are indicative of a decrease in gene expression and values greater than one are indicative of an increase in gene expression. The Figure Legend has been updated with the following statement: 

“Fold change values were calculated by normalizing relative mRNA gene expression (Ct) to GAPDH (�Ct) relative to the control (��Ct). Data is plotted as the fold change of the gene over the conditions (2��Ct).”

4.9. Statistic section is unclear, why did the authors used Paired Student’s t test. And why after ANOVA a pairwise analysis was performed and not a multiple comparison test.

We agree with the reviewer and have revised the statistical tests used throughout the manuscript. The tests used have now been indicated in the Materials and Methods and in the Figure Legends. Where indicated that Student’s t test was used, we now used unpaired. Where indicated that ANOVA was used, we now indicate use of multiple comparison test with Tukey’s analysis.

Results:

1. The images are of very poor quality.

We thank the reviewer for the comment. We have attempted to adjust the quality and clarity of the figures submitted in the revised version of the manuscript including re-graphing the data to and clearly stating the cells used.

2. In some results they refer to the donor number and in others they do not.

We agree with the reviewer that clarification of which cells were used for given experiments was needed. We have now indicated use of either the U87MG astrocytic cell line or any of three primary fetal astrocyte (PFA) donors (Donor 1, Donor 2, Donor 3) on each figure. We have also clarified which cells were used in each relevant section of the Results and in the Figure Legends. Finally, the source of the cells has been clarified in the Materials and Methods.

3. The luciferase assay are confusing, perhaps because it is not described in the methods.

We agree with the reviewer that additional clarification of the luciferase assay is needed. An example of the type of clarifying statements we have included in the manuscript is as follows: To address the ability of Tat to modulate B-catenin activity in physiologically relevant cells, we utilized the U87MG astrocytic cell line and primary fetal astrocyte (PFA) donors in our study. We co-transfected these cells with increasing concentrations of full-length WT Tat and a B-catenin responsive luciferase reporter construct in the presence of LiCl to stimulate B-catenin activity. 24-hours post-transfection, cells were lysed with passive lysis buffer and were incubated with equal volumes of the DualGlo luciferase reagent. Luciferase was measured in a luminometer. Data are presented as B-catenin activity over the experimental conditions.

4. Not all figure legends have the statistical test used.

We agree with the reviewer and have added the appropriate statistical tests to the figure legends.

5. In cases where ANOVA was used, which post-test was performed.

When two-way ANOVA was used it was followed by Tukey’s multiple comparisons test. This is now included in the methods and figure legends.

6. It is inappropriate to use the Student's t to compare more than two groups (example Figure 1)

We agree with the reviewer and have revised the statistical analyses used as indicated in the figure legends. Student’s unpaired t-test is now used to compared two groups only. 

7. The number of experiments is not clear.

 We agree with the reviewer and have added the number of experiments to each figure legend.

The discussion does not have a clear focus and repeats a large part of the results, which makes it tedious and very long.

We agree with the reviewer and have largely revised and rewritten the Discussion for clarity.

---

## [Decision Letter · Decision Letter 1]

4 Mar 2020

Morphine exposure exacerbates HIV-1 Tat driven changes to neuroinflammatory factors in cultured astrocytes

PONE-D-19-16014R1

Dear Dr. Klase,

We are pleased to inform you that your manuscript has been judged scientifically suitable for publication and will be formally accepted for publication once it complies with all outstanding technical requirements.

With kind regards,

Fatah Kashanchi

Academic Editor

PLOS ONE

Additional Editor Comments (optional):

Reviewers' comments:

Reviewer's Responses to Questions

**Comments to the Author**

1. If the authors have adequately addressed your comments raised in a previous round of review and you feel that this manuscript is now acceptable for publication, you may indicate that here to bypass the “Comments to the Author” section, enter your conflict of interest statement in the “Confidential to Editor” section, and submit your "Accept" recommendation.

Reviewer #2: All comments have been addressed

2. Is the manuscript technically sound, and do the data support the conclusions?

Reviewer #2: Yes

3. Has the statistical analysis been performed appropriately and rigorously? 

Reviewer #2: Yes

4. Have the authors made all data underlying the findings in their manuscript fully available?

Reviewer #2: No

5. Is the manuscript presented in an intelligible fashion and written in standard English?

Reviewer #2: Yes

6. Review Comments to the Author

Reviewer #2: The authors responded adequately to the comments, which greatly improved the understanding of the manuscript.

Only one detail remains regarding the qRT-PCR. Figure 6 indicates that the results are 2 ^ ΔΔC and should say 2 ^ -ΔΔCT. Please confirm that the data is calculated correctly. And although it is not mandatory, I suggest that the way in which the analysis was performed must be in the methods section rader than in the figure legend.

7. PLOS authors have the option to publish the peer review history of their article (what does this mean?). If published, this will include your full peer review and any attached files.

Reviewer #2: Yes: Anayansi Molina-Hernández

---

## [Editor Report · Acceptance letter]

11 Mar 2020

PONE-D-19-16014R1 

Morphine exposure exacerbates HIV-1 Tat driven changes to neuroinflammatory factors in cultured astrocytes 

Dear Dr. Klase:

I am pleased to inform you that your manuscript has been deemed suitable for publication in PLOS ONE. Congratulations! Your manuscript is now with our production department. 

With kind regards,

on behalf of

Dr. Fatah Kashanchi 

Academic Editor

PLOS ONE